# Artificial Intelligence Algorithms for Treatment of Diabetes

Mudassir M. Rashid [1], Mohammad Reza Askari [1], Canyu Chen [2], Yueqing Liang [2], Kai Shu [2] and Ali Cinar [1,3,*]

1   Department of Chemical and Biological Engineering, Illinois Institute of Technology, 10 W 33rd St.,
    Chicago, IL 60616, USA
2   Department of Computer Science, Illinois Institute of Technology, 10 W 31st St., Chicago, IL 60616, USA
3   Department of Biomedical Engineering, Illinois Institute of Technology, 3255 S Dearborn St.,
    Chicago, IL 60616, USA
*   Correspondence: cinar@iit.edu

**Abstract:** Artificial intelligence (AI) algorithms can provide actionable insights for clinical decision-making and managing chronic diseases. The treatment and management of complex chronic diseases, such as diabetes, stands to benefit from novel AI algorithms analyzing the frequent real-time streaming data and the occasional medical diagnostics and laboratory test results reported in electronic health records (EHR). Novel algorithms are needed to develop trustworthy, responsible, reliable, and robust AI techniques that can handle the imperfect and imbalanced data of EHRs and inconsistencies or discrepancies with free-living self-reported information. The challenges and applications of AI for two problems in the healthcare domain were explored in this work. First, we introduced novel AI algorithms for EHRs designed to be fair and unbiased while accommodating privacy concerns in predicting treatments and outcomes. Then, we studied the innovative approach of using machine learning to improve automated insulin delivery systems through analyzing real-time information from wearable devices and historical data to identify informative trends and patterns in free-living data. Application examples in the treatment of diabetes demonstrate the benefits of AI tools for medical and health informatics.

**Keywords:** diabetes; digital health; predictive models; artificial intelligence; machine learning; deep learning; automated insulin delivery; electronic health records



## 1. Introduction

Artificial intelligence (AI) has provided new powerful approaches to addressing various problems in medicine [1,2]. This work focuses on machine learning (ML) applications in diagnosing and treating diabetes. Since most ML techniques are data-driven, the source, type, and quality of data play an important role in the performance, accuracy, and reliability of the developed ML models. The data used can be subjective, objective, or a combination of both [3,4]. Subjective sources of data include diaries kept by the individual, her/his responses to questionnaires, and self-reported information such as meals and physical activities entered in various apps that collect such data. Objective data are collected by devices that record measurements of numerous physiological variables and test results reported. Challenges to objective data quality include accuracy of measurements, effects of measurement noise and artifacts, missing values, and outliers in data. Under or overestimation of food consumed, variations in the level of stress or pain perceived by the patient, and forgetting entries to dairies are common in subjective data. When data are collected for various classes of events, the balance in the number of samples in each class can also influence or bias the results of the ML algorithms. This manuscript will first discuss data preprocessing methods to reduce the influence of imbalances and faults in data. Then it will illustrate the use of ML in addressing various challenges in the treatment of diabetes.

Diabetes, a chronic disease that affects one out of every eleven people around the world, will be used to illustrate the ML applications for detection, classification, and

prediction problems. Diabetes has a large impact on the quality of life of people with diabetes and a significant financial burden on individuals and society [5]. Type 1 diabetes (T1D) and Type 2 diabetes (T2D) are two major types of diabetes [6]. T1D is caused by an absolute insulin deficiency, a result of the loss of insulin-producing beta cells of the pancreas. T2D is characterized by the ineffective use of insulin in the body and defects in insulin secretion that lead to relative insulin deficiency in relation to the increased insulin requirements imposed by the insulin-resistant state. While the *cure* of diabetes is the objective of many active research programs, there is no cure approved to date to eliminate diabetes [7]. Many activities focus on improving the *treatment* of diabetes using various classes of pharmacological drugs and medical devices that can help improve the regulation of blood glucose concentrations (BGC) in people with diabetes [8–10].

The role of ML in the classification of the characteristics of T2D, and the detection of the patterns of the daily behavior of people with T1D will be discussed in this publication. The ML problem for T2D pertains to the development of a decision support system that recommends the most effective treatments (drugs prescribed and lifestyle changes) in people with T2D. The ML algorithm can suggest personalized treatment intervention strategies depending on the characteristics of the subject with T2D and the state of the progressive disease, which can lead to either remission (in the early stages of T2D) or treatment of T2D [11–16]. The use of ML in the treatment of T1D focuses on capturing the patterns of the daily behavior of people with T1D [17,18] and to use this information in enhancing the regulation of their BGC and automated insulin delivery [19]. While our focus is on diabetes, the approaches and applications can be extended to many other chronic diseases.

The application of AI algorithms also differs between T1D and T2D, because of distinct timescales and frequencies of decision-making in the treatment of T1D and T2D. T1D necessitates frequent real-time decisions on insulin dosing, which requires AI algorithms to infer patterns from historical data and analyze in real-time multiple streaming data sources. The decisions rendered in T2D typically are on longer-term timescales, with algorithms developed to offer pharmacotherapy suggestions and lifestyle modification recommendations to care providers for more effective personalized treatment of patients. The differences in the frequency of data and the timescales for rendering decisions lead to the development of different ML algorithms for these two types of diabetes.

Training AI algorithms to help diagnose and treat T2D is challenging because of the diverse spectrum of the chronic disease across the population, the various treatment options and alternatives, and the potential existence of comorbidities and diabetes-related complications. The complex pathophysiology of T2D necessitated the development of different classes of oral and injectable drugs that target specific biochemical pathways to address various characteristics associated with T2D [20,21]. Treatment can include the administration of one or more of these drugs [22,23]. In case of comorbidities, such as cardiovascular diseases, chronic kidney disease, or diabetes complications such as nerve damage, foot complications, and eye disease, additional medications to treat these conditions are also administered. Drug-drug interactions and adverse side effects are common. Lifestyle, including diet and physical activity, is a critical component of the treatment of T2D. Physical activity can increase the risk of hypoglycemia with certain anti-diabetic medications. Thus, the diet, activity level, and lifestyle of the patient must be taken into consideration. A major goal of ML systems that assist medical professionals to personalize treatment approaches is to determine the characteristics and state of T2D for a subject, classify the subject in the most likely cluster of patients with similar characteristics [16,24] and identify the treatment options that have the highest probability of success for this individual [16].

Electronic health records (EHR) capture the medical records of people with T2D and provide a good source of data for clustering people with similar characteristics and identifying treatment(s) that have been successful for each cluster. This data is heterogeneous, including free text, structured information (e.g., entered from hierarchical drop-down

menus) and numerical values (e.g., lab results), necessitating natural language processing (NLP), planning, and structuring for analysis. ML can also identify secondary connections to undiagnosed or potential comorbidities. AI will determine the interventions that are most likely to be effective for specific clusters of patients with T2D. Interventions include various types of antidiabetic drugs that affect different aspects of the metabolism and have different side effects and lifestyle changes (diet, physical activity, stress, and sleep). Currently, the management of all these factors, including the selection of the specific drugs and drug doses that would be appropriate for an individual relies on the experience and decisions of the medical care provider. We will pull the diagnosis codes, laboratory data, medications, and visit types for patients over a 10-year period. The data will include the free text notes from primary care physicians, nurses, social workers, and nutrition notes. Biomedical informatics can leverage ML algorithms to gain insights from historical EHR systems.

The first step in EHR data mining is identifying missing data and errors, followed by data reconciliation. Then, concerns on bias in data and data size will be addressed. Cinar and Rashid have developed various techniques for detecting and reconciling missing data and errors by multivariate statistical and systems engineering techniques [25–27]. A different set of ML approaches have been developed by Shu (fair and robust AI, learning with weak data) [28–32]. Bias in data results from patient records in EHR not reflecting the population and causes models and relations that disproportionally concentrate on the characteristics of the dominant groups in EHR data. Even if EHR has good representation of African Americans and Latino/Hispanics, bias can still creep in unless this data is used with care. The data size provides a different challenge. For example, the accuracy of deep learning with long-short-term memory neural networks improves with large sets of data, but other techniques such as support vector machines, k-nearest neighbors, decision trees, linear discrimination, and ensemble learning provide good results with smaller data sets.

The outcome of the AI techniques for classification and suggestion of the most effective treatments must be trustworthy. Since EHR data may be fragile, clinicians would like clarity on the reasons for the selection of a treatment [33,34]. Explainable AI methods [35–37] provide insight into the selection of interventions in terms that are familiar to care providers. We will focus on explainable AI techniques based on feature attribution methods that assign weights to features used to predict the best treatment [38–40]. The goal is to assign a score to quantify the importance of features that are more useful for the prediction results using attention mechanisms [30]. Another type of explainable ML models aims to derive explanations after the model is trained, or post-hoc explanations [39].

In contrast to the infrequent decision-making in T2D, AI algorithms for T1D must process both historical records and real-time streaming data to make numerous recurrent decisions. People with T1D must make over a hundred decisions every day to regulate their BGC based on complicated calculations on their insulin dose requirements throughout daily life. This complex decision-making must consider numerous factors such as meal carbohydrate amounts and times, past insulin infusions, and planned exercise and future schedules. Assimilating diverse sources of information and making the insulin dosing decisions, and decisions on food types and amount to consume throughout the day, in order to maintain tight glycemic control is a challenging task. Automated insulin delivery (AID) is an advanced treatment approach for keeping BGC in range despite major disturbances to glycemic dynamics such as food consumption and physical activities (PA) [41–51]. Commercially available AID systems use BGC reported every 5 min (based on subcutaneous glucose concentrations measured by continuous glucose measurement [CGM] systems), a control algorithm that computes the optimal insulin dose to be infused, and an insulin pump that infuses the computed insulin dose to subcutaneous tissue [52,53]. These AID systems are called hybrid closed-loop systems because the user manually provides information about the food consumed and then makes adjustments to the target BGC values or insulin dosing to mitigate the effects of routine, scheduled PA. They have achieved 65–75 percent time in range for BGC in clinical trials reported to regulatory agencies [54–59]. However,

some people prefer fully-automated AID systems or would like to be notified when they have not communicated a disturbance to their AID. This missing information may be due to the inability of a young child or a geriatric person with T1D to enter such information, forgetting to enter the information (such as a meal consumed) or an unplanned/unexpected PA (such as running to catch a bus).

Fully-automated AID systems (Figure 1) necessitate the detection of food consumption and estimation of the carbohydrate content of the meal and the detection of PA, classification of its type (such as aerobic, resistance, or interval training), and estimating its intensity and duration [60–64]. This can only be achieved by a multivariable approach where physiological data from wearable devices such as wristbands are integrated with CGM and insulin infusion information to capture the metabolic state of the individual and compute the optimal insulin dose to be infused by the pump [62]. The model predictive control (MPC) approach has been demonstrated as the most effective type of control algorithm for AID systems [26,65,66]. It necessitates the estimation of the future BGC to determine the optimal insulin dose to be infused by the pump. The accuracy of future BGC estimates increases when future disturbances are estimated and included in predicting future BGC excursions [19]. ML algorithms can analyze glucose and insulin data to determine the insulin dose requirements and analyze historical data to identify trends and patterns that can be used to anticipate events and proactively make insulin dose adjustments [19].

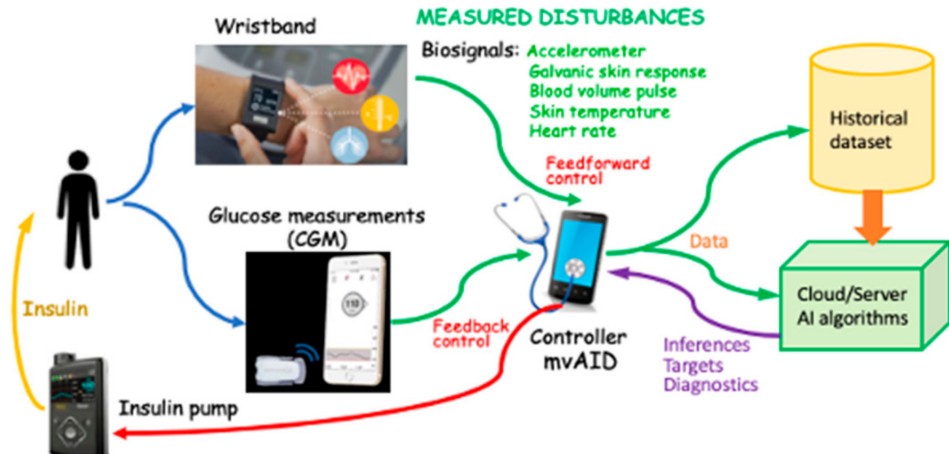

**Figure 1.** Devices and information flow in multivariable automated insulin delivery systems. Historical data set depository and cloud/server for AI algorithms are used to improve glucose control by incorporating information on identified historical patterns.

Unfortunately, both interpersonal variations in BGC dynamics among people with T1D and intrapersonal variations in BGC dynamics of an individual throughout a day, reduce the accuracy of fixed models developed based on historical data. Consequently, recursive models to predict future BGC trends and adaptive control techniques become more appealing for use in AID systems [27,62,66]. Their performance improves further by providing information from ML algorithms to detect or predict disturbances that affect BGC dynamics.

AI algorithms are gaining prominence in automating complex tasks in the management of diabetes. The paper is structured to introduce various methods to address the challenges in improving detection, classification, and prediction accuracy for better treatment of T2D and T1D in the Methods section. ML for better analysis of electronic health records (EHR) of people with diabetes will be introduced in Section 2.1. Section 2.2 focuses on ML techniques integrated with data preprocessing, imputation of missing data, and data reconciliation to predict the occurrence of food intake and exercise, and their concurrent incidence. Section 2.3 presents the integration of ML with automatic control to improve BGC regulation with AID. Section 3 presents the results of implementing the ML algorithms

presented in Section 2.2 to data collected in free-living, and simulations to illustrate the effects of ML algorithms outlined in Section 2.3. Section 4 provides the discussion of results, and Conclusions are given in Section 5.

## 2. Methods

In this section, we first discuss the use of fair and trustworthy AI algorithms trained using EHRs for personalized medicine and clinical decision-making in T2D. Then, we investigate novel methods to analyze free-living data from wearable devices for use in mobile health and diabetes technologies for the treatment of T1D.

### 2.1. ML with Electronic Health Records for Personalized Medicine and Decision Making in T2D

EHR data can be generally divided into two categories: structured data and unstructured data. Structured EHR data includes demographic information, laboratory results, prescribed medication, diagnosis codes, etc. Unstructured EHR data contains the clinical notes and free text notes written by doctors and medical practitioners. ML models on both types of data have been shown biased against certain demographic groups [67]. For example, the predictions of multiple neural models on the MIMIC-IV dataset have been found to have race-level discrepancies [68]. There exists a performance disparity between demographic groups for a gradient boosting model on the Translational Research Integrated Database Environment (STRIDE 8) dataset [69]. The use of pooled cohort equations (PCE) method is also verified as biased to help guide physicians [3,70]. In addition to the structured data, bias issues also exist in unstructured data for tasks such as automated question answering (QA) tasks [71] and text generation [72].

The goal of general equitable/fair ML is to build an effective algorithm for prediction while still satisfying fairness constraints. Following existing fair ML work [73,74], the performance of fairness can be measured with metrics such as equal opportunity and statistical parity. Equal opportunity requires that the probability of positive instances with arbitrary protected attributes being assigned to a positive outcome are equal; statistical parity requires the behavior of the prediction model to be fair to different sensitive groups. Bias in algorithms can come from different sources such as unrepresentative or incomplete training data or the reliance on flawed information that reflects historical inequalities [75]. Bias can also come from the algorithm design without considering fairness criteria or come from user interactions such as user behavior bias.

To ensure fairness in ML algorithms, recent advancements in fairness-aware models aim to optimize different fairness measures including individual fairness and group fairness: individual fairness aims to give similar predictions to similar individuals with counterfactual fairness or fairness through awareness; while group fairness is to treat different groups equally measures like statistical parity and equalized odds.

An algorithm that is unbiased to the sensitive attributes, $s \in \{0, 1\}$, can be obtained by incorporating constraints in the optimization problem that impose fairness on the model. The goal of finding a mapping that provides both accurate predictions and fairness guarantees can be achieved by incorporating constraints that ensure the parameter estimates result in classification results that are independent of the sensitive attributes.

The model parameters of a fair model can be obtained by solving the following constrained optimization problem:

$$\theta^* = \operatorname*{argmin}_{\theta \in \Theta} \mathcal{L}(\theta, \mathcal{D}) \tag{1}$$

$$\text{s.t.} \quad C(\theta, \mathcal{D}) \le \delta \tag{2}$$

with

$$C(\theta, \mathcal{D}) := \max_{y \in \{0,1\}} \left[ P(\hat{y} \ne y | s = 0) - P(\hat{y} \ne y | s = 1) \right] \le \delta \tag{3}$$

where a relaxed probabilistic constraint computed over the training data $\mathcal{D}$ is introduced so that the estimated model parameters yield approximately equalized odds, biases, and accuracies with respect to the protected attribute $s$ and outcome $y$. The constraint, $C(\theta, \mathcal{D})$, expressed generally as $P(\hat{y}|x, s) = P(\hat{y}|x)$, or the independence of the predicted outputs with respect to the sensitive attributes, enforces conditional independence between model predictions $\hat{y} = f_\theta(\mathcal{X})$ and the sensitive attribute $s$ as conditional prediction parity [76,77], ensuring the overall misclassification rate is not affected by the sensitive attributes. The constraint may also be formulated in other analogous forms [73], such as false negative rate, false positive rate, false omission rate, and false discovery rate [74].

Based on which stage of the ML training processes are considered, existing approaches fall into three categories: pre-processing approaches, in-processing approaches, and post-processing approaches [78,79]. The pre-processing approaches are applied before training machine learning models by modifying the labels and/or attributes in the training data; For example, one can reweight the datasets to be less biased to sensitive attributes before feeding them into an algorithm. In-processing approaches aim to incorporate constraints during model training to ensure fairness. For example, adversarial debiasing aims to train a fair algorithm with adversarial learning to minimize the discrepancy between different sensitive groups, and post-processing methods develop mechanisms to change predicted labels. For example, one can calibrate the predictions based on fairness criteria such as Equalized odds. It is important to consider privacy when building a fair ML model on EHR data as it is often highly confidential due to legal compliance.

Apart from ensuring fairness in various medical applications, it is also essential to preserve privacy in handling all kinds of medical data. Privacy protection is one of the core components in jurisdictions on healthcare data across the globe. Ensuring privacy in ML models for healthcare applications has been an increasingly critical concern. ML models can have serious privacy leakage issues for healthcare information. For example, a study shows that ML models can be utilized to identify the individuals in healthcare data repositories despite the removal of protected sensitive information [80]. Therefore, multiple privacy protection methods including differential privacy, local differential privacy, and federated learning can be incorporated to develop privacy-preserving ML models with healthcare data. For instance, differential private logistic regression and naive Bayes are proposed for breast cancer classification and diabetes prediction problems [81]. Federated learning is applied for heart failure disease prediction in a private manner [82]. Thus, it is important to study how to ensure fairness under privacy. Due to the restrictions such as Electronic Communications Privacy Act (ECPA), General Data Protection Regulation (GDPR) and medical data requirements, it is usually infeasible to directly get access to the original sensitive data and effective privacy protection methods are needed [83].

Various privacy protection approaches are proposed to protect the privacy of sensitive data including multiparty computation (MC), differential privacy (DP), local differential privacy (LDP) and federated learning (FL). Such privacy mechanisms are widely deployed in medical analysis applications. There is currently limited research on the intersection of privacy and fairness in ML. For example, with regards to the problem of fairness under LDP, we can observe the experiment results on two mainstream fairness datasets (ADULT [84], COMPAS [85,86]) and conclude that stronger privacy guarantee requirements lead to worse fairness performance for debiasing models such as adversarial debiasing (Figure 2). Therefore, there is a trade-off between the debiasing model and the LDP privacy mechanism. More research is needed for the direction of private and fair ML.

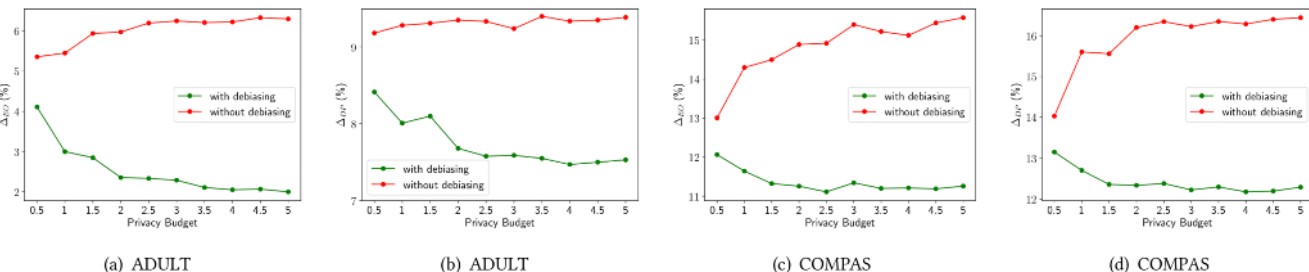

**Figure 2.** Assessing the impact of fairness performances under private data. (**a**,**b**) ADULT, (**c**,**d**) COMPAS [87].

In addition to privacy, we also propose to deal with the scenario when we have limited/unknown sensitive attributes (e.g., gender, race) to ensure fairness in ML models. In real-world medical systems, user-sensitive attributes are often limited or even unavailable due to legal restrictions or security protection. In [31], we study the problem of learning fair classifiers by exploring related attributes. Since the related features might also be useful for classification, instead of simply discarding them to improve fairness, we propose a novel framework that simultaneously utilizes them to learn fair classifiers and adjusts the importance weight of each related feature to trade-off their contribution to classification accuracy and fairness. An illustration of the proposed FairRF is shown in Figure 3. It is mainly composed of three parts: (i) a base classifier that predicts the label of a data sample; (ii) a covariance regularizer, which utilizes each feature to alleviate bias; and (iii) an importance weight learner, which trade-offs the contribution of each related feature for classifier accuracy and fairness. The overall loss is to minimize the prediction error $L_{cls}$ and the weighted sum of correlation regularization from $K$ related attributes $L_f$. Experimental results on the mainstream fairness datasets (ADULT, COMPAS) show that FairRF can significantly improve the performance of fairness metrics such as Equal Opportunity with 10–40%, without losing much on classification accuracy.

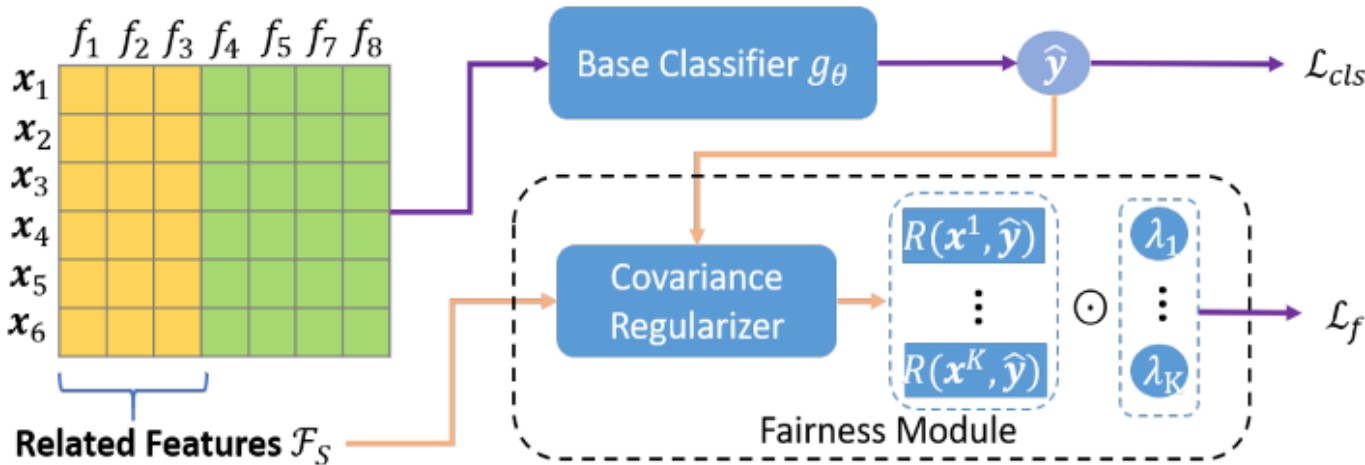

**Figure 3.** Fair classification with related sensitive attributes [31].

In addition to learning from related attributes to estimate sensitive attributes for fair classification, we may explore auxiliary information from external sources to help. Specifically, when there are no sensitive attributes for training a fair model in the target domain, there might exist a similar domain that has sensitive attributes. For example, when predicting T2D with data that is collected recently when it is restricted from accessing sensitive attributes, it is possible that abundant historical data is available from the past. In ML, domain adaptation [88–90] techniques have been investigated for transferring knowledge across domains for classification; however, they cannot be directly applied to fair classification. We propose to investigate an unsupervised domain adaptation framework

to transfer fairness without sensitive attributes in the target domain [91]. We will use retrospective EHR datasets previously obtained from the University of Illinois Hospital and Health Sciences System (UI Health). The EHR dataset consists of approximately 300,000 adult patients treated at UI Health over the course of a decade in duration, with at least two visits in a 24-month time span, and either diagnosed with T2D or no diabetes. The proposed framework is to investigate adversarial learning for joint debiasing and adaptation. First, in the adversarial debiasing, the optimization is to predict the labels (e.g., diabetes) accurately while not accurately assessing user-sensitive attributes; Second, in the adversarial domain adaptation, the goal is to learn an accurate function to estimate sensitive attributes in the target domain. More research will be investigated to deeply explore the effectiveness of the above framework in EHR data with respect to different domain variations (e.g., quantify the similarity between domains that are categorized according to time, location, etc.).

### 2.2. ML with Free-Living Data for Digital Health Technologies in T1D

In people with T1D, the inability of the body to produce insulin necessitates exogenous insulin administration to regulate the BGC. The advent of CGM devices enabled people with diabetes to monitor their BGC every five minutes and interpret the numerical and graphical information displayed. People with T1D can administer insulin by injections by using injectors or insulin pens or by infusions with insulin pumps to the subcutaneous tissue. While these technological advances improved BGC regulation and insulin delivery, estimating the correct amount of insulin to administer is still a major burden for people with T1D and their care providers. People with T1D make over 100 decisions every day, distracting them from their other activities, increasing the probability of making incorrect decisions, and reminding them of their chronic condition continuously.

People have patterns in their behavior that may vary between workdays and weekends, during seasons of the year, and on holidays. Capturing these patterns, in particular meal and physical activity information, provides useful information and enables suggestions for insulin boluses, reminders, and modifications in the target levels of BGC. They can also be used by AID systems (also called artificial pancreas).

Determining the habits and behavior patterns of individuals with T1D can increase the time-in-range (keeping BGC in the range of 70–180 mg/dL) and minimize hypoglycemia (BGC < 70 mg/dL) and hyperglycemia (BGC > 180 mg/dL) episodes of people with T1D in free-living. The data used for pattern detection can include BGC measurements reported by CGM devices that measure subcutaneous glucose concentrations, insulin infusions reported by the insulin pump, physiological data reported by wearable devices such as wristbands, and information provided by the individuals, such as meal consumption and amount, and the beginning time and duration of physical exercise. ML with free-living data has many challenges: imbalance in data introduces bias in the classification of daily events such as meal consumption and physical activity. Furthermore, missing values, sensor noise, and outliers reduce the accuracy of ML results. Some of the missing information is caused by interruptions in communication between devices and others by forgetting to enter manual information by the individual with T1D. Hence, data must be preprocessed to reconcile the effects of artifacts, measurement noise, and outliers and impute missing values.

The probabilities of meals, physical activity events, or their concurrent presence during one day (24-h period) were obtained by analyzing 15 months of CGM, insulin pump, meal, and physical activity data collected from a randomly selected person with T1D from the Tidepool dataset (Figure 4) illustrate that while there are peak times for meals and exercise, their high probabilities of occurrence are spread over time windows ranging from 1 to 3 h.

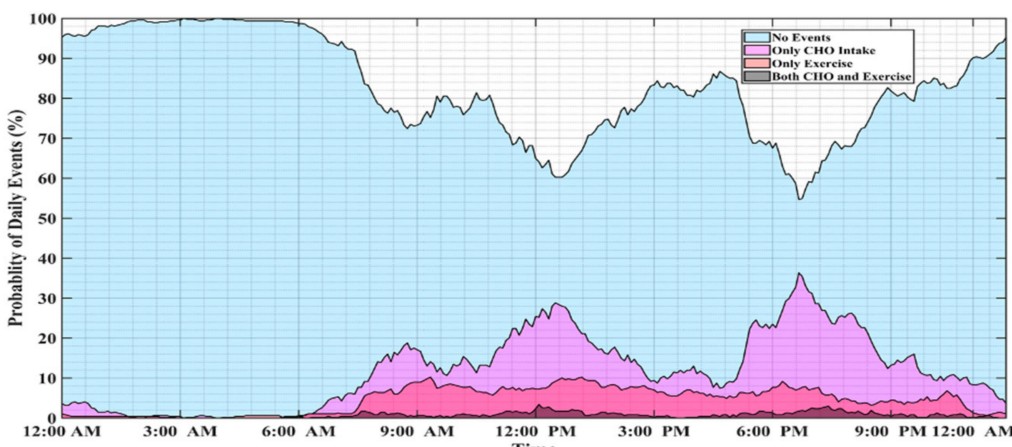

**Figure 4.** The probability of meal and physical activity events during a day was obtained by analyzing 15 months of the pump-CGM sensor, meal, and physical activity data of a random person with T1D from the Tidepool dataset [18].

The Tidepool dataset is used to illustrate various data preprocessing activities and the efficacy of ML for detecting and discriminating meal and exercise events (including their concurrent occurrence). It includes data from 300 self-collected T1D datasets from people who donated their data for research and the de-identified data were made available for research. Each dataset represents data from one individual with T1D. 50 datasets in this database include CGM sensor-insulin pump recordings and exercise information such as the time, type, and duration of physical activity recorded from either open/closed-loop insulin pump-sensor data. Meal information is reported as the amount of carbohydrates (CHO) consumed in the meal as estimated by the subject (over or underestimation of CHO in meals is common). The subjects with T1D selected for this study used insulin pump-CGM sensor therapy for up to two years, and some of them have lived with diabetes for more than fifty years. The details of the demographic information of the selected subjects in our work and the definition of the variables collected are provided in [18]. Separate models are developed for each person in order to capture personalized patterns of meal consumption and physical activity. Data lengths vary between a minimum of 30 days up to 700 days of recorded information. The demographic information of the donors is provided in [18]. The donors have used CGM data for adjusting their insulin dosing (by using sensor-augmented-pump or automated insulin delivery) therapy for up to two years. All information except meals and physical activity are recorded every five minutes. As expected, in free-living there are many missing values in data and signal noise that must be addressed by preprocessing the data. The probabilities of a meal or physical activity events or their concurrent presence during one day (24-h period) were obtained by analyzing 15 months of CGM, insulin pump, meal, and physical activity data collected from a randomly selected person with T1D from the Tidepool dataset (Figure 4) illustrate the high probabilities of occurrence of these events spread over time windows ranging from 1 to 3 h, and the peak times for meals and exercise. It will be the task of a ML algorithms to detect and discriminate the daily patterns of behavior for each person.

In data preprocessing, data reconciliation, outlier removal, and imputation of missing values are conducted using effective algorithms. The algorithms developed to implement these tasks are reported in [18]. The diagram of workflow for data processing, feature extraction, and modeling is summarized in Figure 5.

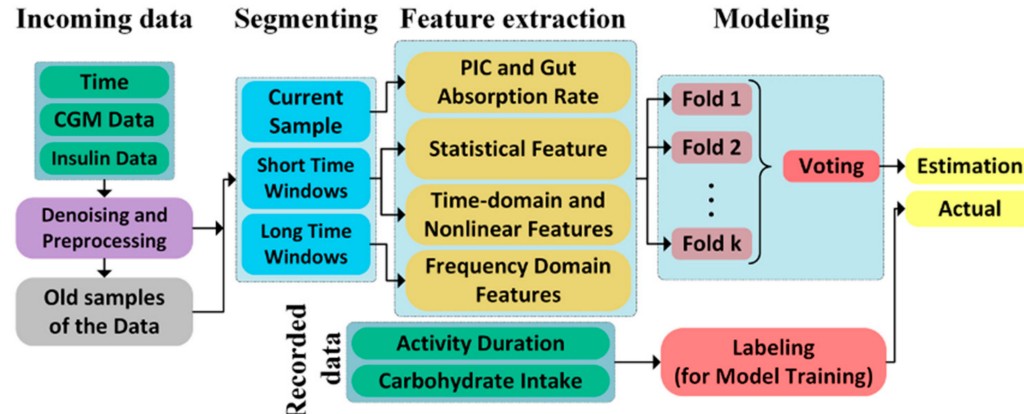

**Figure 5.** The diagram of workflow for data processing, feature extraction, and modeling [18].

For the detection and classification of events, the CGM values and insulin pump data are preprocessed and stacked with the past recorded data. All samples are segmented into three different time windows: the current samples are used for model-based feature extraction, two-hour window of the past data are used for statistical and nonlinear feature variables, and past day samples of the data are used for the frequency domain features. Redundant feature variables are identified and eliminated by a two-step feature selection procedure. First, the deviance statistic test is performed to filter out features with low significance. Then, the training split of all datasets is used in the wrapper feature selection strategy to maximize the accuracy of the classifier in estimating meal and exercise events. Voting and k-fold cross validation are used to assign the current state to one of the four classes: "Meal and Exercise" (M and E), "no Meal but Exercise" (E), "no Exercise but Meal" (M), "neither Meal nor Exercise" (No M and E).

Event detection is achieved with recurrent neural networks (RNN). RNN models are developed for each individual to capture their behavior more accurately. The RNN models are used in this work to handle the temporal sequential nature of the time-series data, where memory in the network ensures that the current state can effectively influence the future state after a certain allotted time interval. Several RNN model configurations are studied to assess their accuracy and performance in estimating the probabilities of occurrence of these four classes at each sampling time [18]. The feature variables are time-series data, and recorded samples of past values of all feature variables are stacked together to build the tensor of model inputs. These models use a past window of two hours, corresponding to 24 past samples of the recorded CGM and insulin pump data, and extracted features. Event (M, E) estimations are performed one sampling time backward. The schematic diagram of the most promising RNN structure is illustrated in Figure 6.

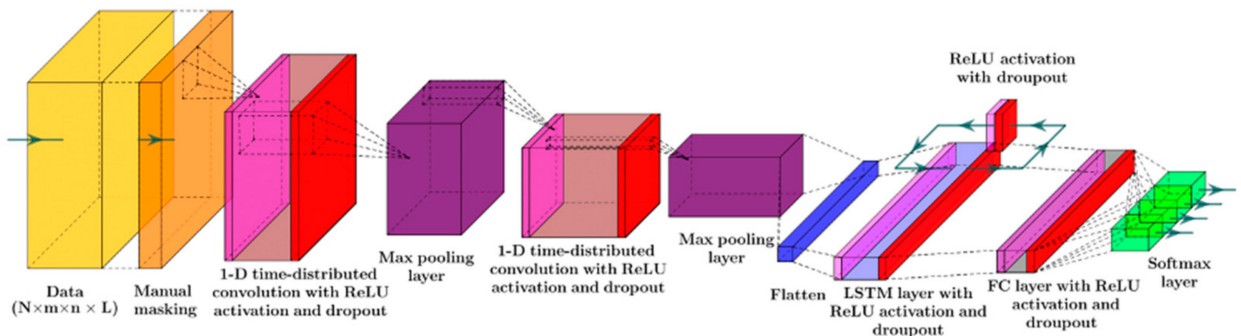

**Figure 6.** 1-D Convolutional RNN with LSTM layer [18].

### 2.3. Merging ML with Automatic Control to Improve Glucose Regulation

People with diabetes have mental models of their own metabolism and can provide additional information to their AID systems manually that enable the algorithms used in automated decisions to improve the insulin dose computed. The most prevalent manual information announces potential impending glucose excursions during meals and physical activity, including user-reported meal information for computation of an insulin bolus and increase in the target BGC and reduction of basal insulin infusion before starting an exercise session. The first generation of AID systems automated the manual activities that were performed repeatedly by the user [92]. Various control algorithms were proposed over the years for AID systems, and eventually converged on using model predictive control (MPC) systems. In MPC, a dynamic model representing the metabolism of the user forecasts the future behavior of the BGC based on hypothesized future control actions that are optimized in real-time to bring future BGC as close to its desired trajectory as possible, despite various disturbances such as meals and physical activities.

The second generation of AID systems aims to move from hybrid to automated operation and reduce manual inputs to a few urgent situations [52,53]. This can be achieved by measuring and interpreting additional variables in real-time to infer the presence and properties of disturbances to glucose homeostasis. Further, the expected behavior of a person can be modeled using historical data sets and the model can be used to improve insulin dosing decisions. This multivariable AID (mvAID) captures additional information from a wearable device (i.e., blood volume pulse, heart rate, electrodermal activity, accelerometer data) to provide real-time information to AID. Such information can be interpreted by ML models to infer the presence of physical activities, classify their type, estimate their intensity, and predict their effects on glucose levels in the near future [19,62,66,93–95].

While the current mvAID with physiological data streaming in real-time improves BGC regulation and ML of personal habits enhances BGG regulation, it does not leverage the full promise of AI. It has limited capability to address the challenges created by the human in the loop, where the user who has access to data and information such as a potential hypoglycemia episode changes their behavior and reduces the accuracy of the BGC predictions and control decisions (insulin doses) made by the mvAID system (Figure 7). MPC uses the predicted (future) BGC values to compare them against the desired future BGC levels and conducts optimization in real-time to bring the predicted BGC values to the desired (target) values by modifying future insulin doses. The use of the daily behavior patterns captured by ML improves the model prediction accuracy of the used MPC and updates the parameters of the MPC for tighter regulation of BGC. The RNN models developed in the previous section can capture these patterns and provide information to the MPC on the specific pattern that is evolving on a given day in order to predict future disturbances more accurately.

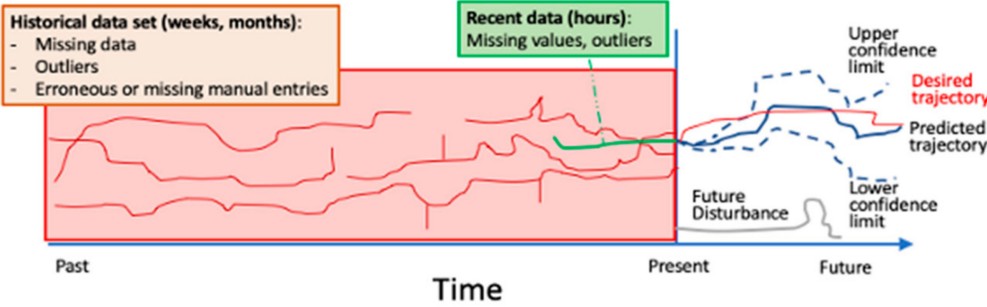

**Figure 7.** Lengths of historical data to capture many behavior patterns of a person (red), recent past data for use by MPC (green) and challenges in data missing values, outliers). Confidence limits in the prediction of future BGC, and disturbance effects.

## 3. Results

The results for the 1-D convolutional RNN with LSTM layer for the detection and classification of the four states for one subject (Subject 2) in terms of Total Accuracy (%), Weighted Recall (%), Precision (%), and F1 Score (%) are 94.69 (0.33), 94.69 (0.33), 96.58 (0.20), and 95.31 (0.16), respectively, where the standard deviations are given in parentheses. Some erroneous detections, such as confusing meals and exercise are dangerous since meals necessitate an insulin bolus while exercise lowers BGC, and elimination of insulin infusion and/or increase in target BGC are needed. RNNs with LSTM and 1D convolution layers provide the best overall performance in minimizing such confusions: two meals events are classified as exercise (0.003%) and eight exercise events are classified as meals (0.125%). The performance of the four different RNNs are illustrated for one person in the Tidepool dataset in Figure 8.

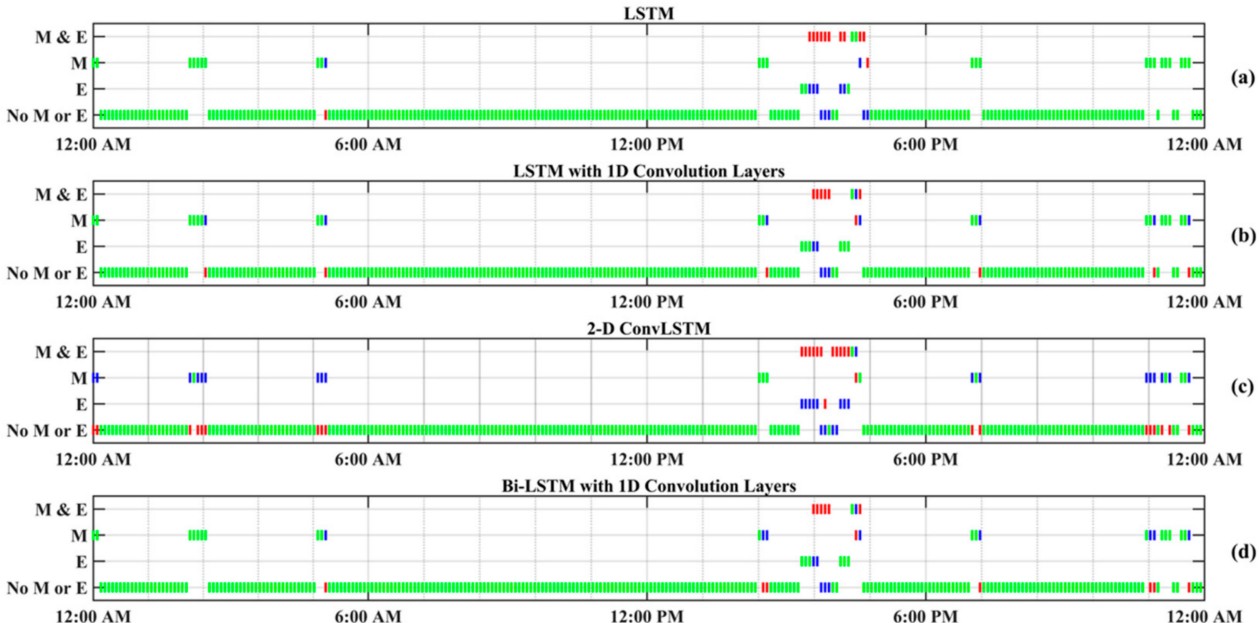

**Figure 8.** Meal and exercise events were predicted for one person (subject 2) with outliers and unimputed samples by using different RNN configurations. Vertical green bars represent correctly predicted classes. Vertical red bars denote incorrectly predicted classes, and their actual labels are shown by blue bars. (**a**) LSTM, (**b**) LSTM with 1D Convolution Layers, (**c**) 2DConvLSTM, (**d**) Bi-LSTM with 1D Convolution Layers [18].

Figure 8 displays a random day selected from the test data to compare the effectiveness of various RNN models in detecting meal and exercise disturbances. Among four possible realizations for the occurrence of events, detecting joint events, Class M and E is more challenging as it usually shows overlaps with Class M and Class E. Several reasons contribute to this challenge. Many people with T1D usually have a small snack before and/or after exercise sessions. Also, exercise alone and exercise and snack together are less frequent than sedentary state and meal consumption, and consequently, there is less information on Class M and E. Furthermore, the AID systems used by subjects record automatically only CGM and insulin infusion values, and meal and physical activity sessions needed to be manually entered into the device, at times an action that may be forgotten by the subject.

Meal consumption and physical activity are two prominent disturbances that disrupt BGC regulation, but their opposite effect on BGC makes the prediction of Class M and E less critical than only meal or physical activity classes. The confusion matrices of the classification results for one of the subjects [18] and Figure 9 indicate that detecting Class E is slightly more challenging in comparison to Class E and Class no M and E. One reason for this difficulty is the lack of biosignal information such as 3D accelerometer, blood volume

pulse, and heart rate data in the Tidepool data set. As the use of wearable devices increases, this difference is expected to be reduced [96–98].

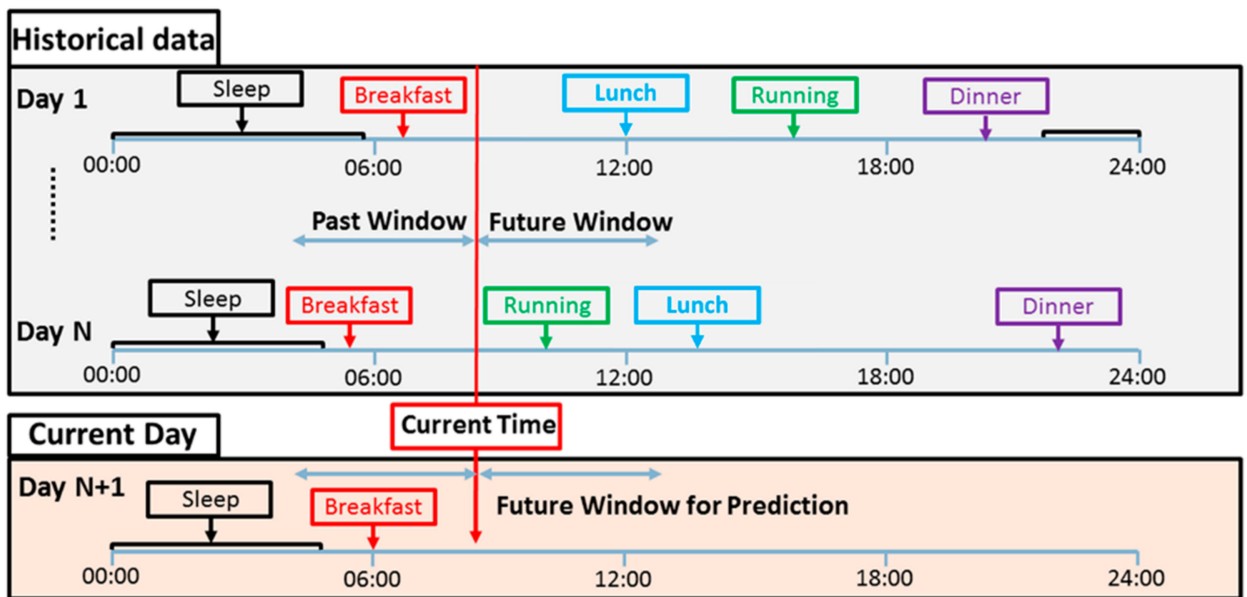

**Figure 9.** Disturbance prediction from historical data [19].

Following the detection of meal and exercise events in historical data, we combine the knowledge in historical data with real-time streaming information to determine the most likely behavior pattern for the subject, and incorporate the estimated behavior pattern in the multivariable AID system for improved decision-making. This improvement is illustrated by simulations conducted with the mvAID connected to a multivariable simulator (mGIPsim) [99]. mGIPsim is a novel software enabling the in silico testing of advanced mvAID systems that improve glycemic control by supplementing the CGM data with additional physiological measurements. The mGIPsim software has descriptions of meals, insulin administration, and physical activities as inputs, and provides glucose and insulin concentrations and physiological variables as outputs for use by nvAID algorithms. mGIPsim can simulate patient behavior for up to one year with random daily variations in meal and physical activity characteristics within ranges specified by the user. It has a heterogenous virtual synthetic patient population that is representative of the actual patient population. mGIPsim is used to generate data for several days in response to daily meal and exercise events. This historical data set is used to extract patterns of behavior for each virtual subject. On the current day, the data generated up to the current time is compared with all patterns of behavior of that subject, and the pattern that matches the most to the current day is selected to provide future meal and exercise information (Figure 9). This pattern information is used by the MPC to incorporate future potential meal and exercise events in the prediction of future glucose values and compute optimal insulin infusion to minimize the deviation of the predicted glucose concentration from the desired glucose concentration.

The improvement achieved by this approach is illustrated by two figures that compare the performance of the AID control systems. The mvAID with an adaptive glucose concentration model is denoted by A-MPC, and the controller that learns the patterns of behavior is denoted by AL-MPC in Figures 10 and 11. The first figure (Figure 10) compares the performances on day 30 of the simulation period for the two algorithms, by reporting the mean glucose concentrations and their standard deviations for all 20 virtual subjects simulated. AL-MPC is able to reduce the maximum glucose concentrations without increasing the risk for hypoglycemia. Figure 11 illustrates the performance of the two algorithms for one subject over the 30-day simulation period. Again, AL-MPC is able to reduce the

maximum glucose concentrations without increasing the risk for hypoglycemia. The number of predicted hypoglycemic events decreased for the 30-day simulation period from 92 to 75 with AL-MPC (hypoglycemia did not occur since based on the prediction rescue carbs are consumed by the subjects) and the average of total daily injected insulin (U) remained the same (39.5 U) [19].

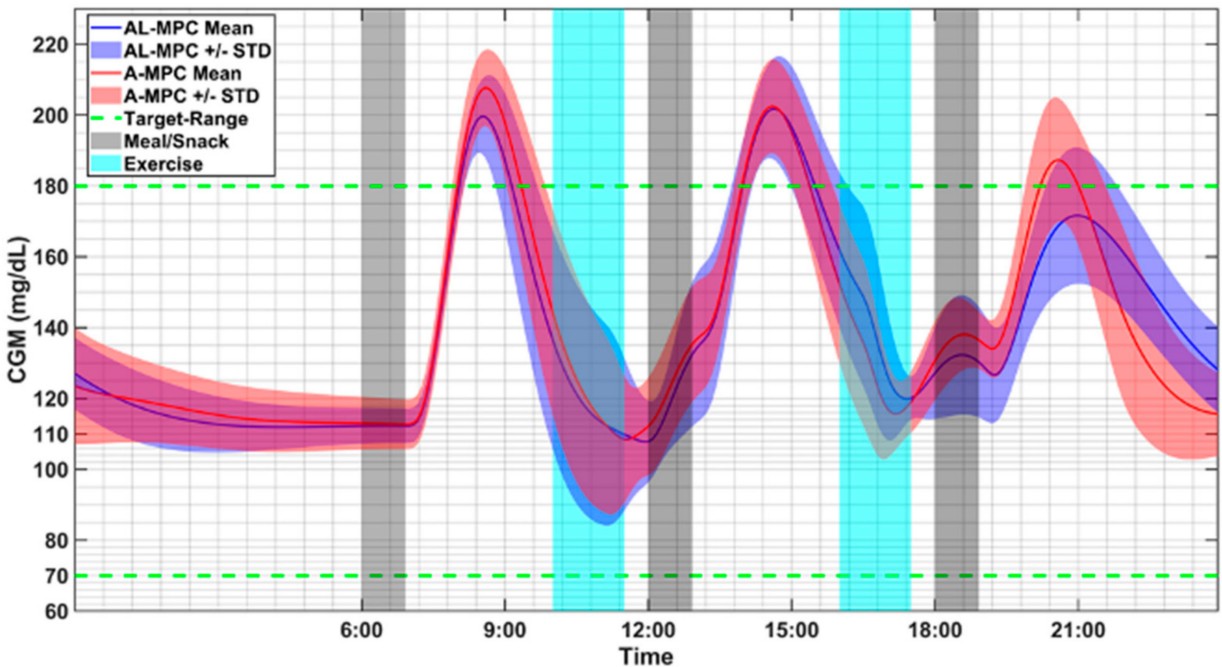

**Figure 10.** Closed-loop results of mAID for all subjects on day 30 [19].

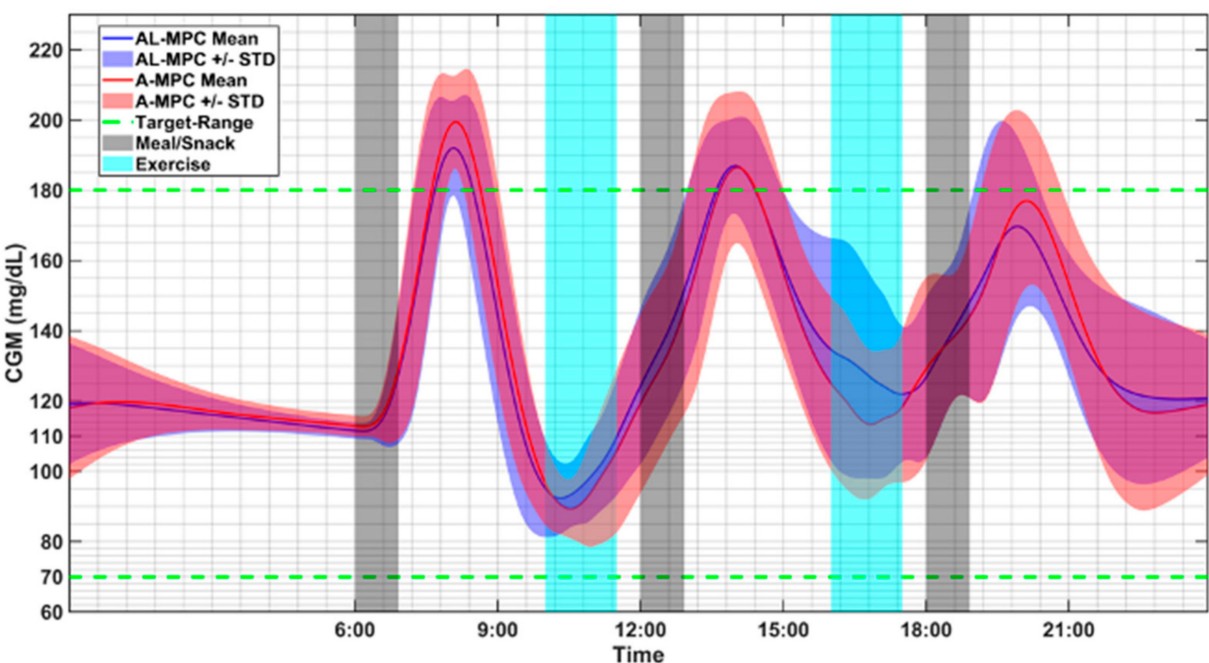

**Figure 11.** Closed-loop results of mAID for one select subject during whole simulation [19].

## 4. Discussion of Results

The applications of ML and AI in healthcare applications, particularly the treatment of diabetes, are explored in this work. The use of various approaches to enhance AI to

diagnose the characteristics and state of a person with T2D using EHR data is explored. Methods are discussed to ensure fairness and privacy of the AI algorithms in critical areas such as healthcare applications. The use of reinforcement learning enabled complex mathematical programming problems to be solved, enabling the optimization formulations for model learning to explicitly consider properties for fairness and privacy.

In contrast to the application of AI in T2D where EHR data are only updated occasionally upon the availability of new laboratory results and doctor assessments, the use of AI in T1D necessitates frequent rendering of insulin dosing decisions based on both historical information and real-time streaming data. We analyzed historical data from a large dataset to identify the likelihood of meal and exercise events, which are significant disturbances to the tight regulation of BGC in diabetes. We showed that the meal and exercise events can be predicted with high accuracy using a 1-D convolutional RNN with LSTM.

The incorporation of behavioral patterns discerned from historical records in the real-time insulin dosing decisions is shown to significantly improve the BGC regulation in people with T1D. The advanced knowledge gained by analyzing the data from the current day to estimate the likely behavioral pattern and anticipate the future pattern of the subject enables proactive mvAID systems that can mitigate the effects of disturbances to tight BGC control before the disturbance effects materialize.

The applications of ML and AI in medicine show promising results in transforming the treatment of chronic diseases such as diabetes. As the AI algorithms improve, they will be able to better identify clusters, patterns, and relations in the medical data. The AI algorithms will reduce the burden of managing chronic diseases and enhance the quality of life of patients.

## 5. Conclusions

The developments in ML, EHR systems, wearable devices, and medical testing provide a fertile environment for powerful detection, classification, modeling, monitoring, and control applications in medicine. Increased adoption and deployment of AI in medicine necessitates advances in reducing bias, reconciling missing values and outliers to make accurate and precise analysis of data, and building trustworthy and responsible AI-based systems for medical applications. The advancements in trustworthy, responsible, robust, and reliable AI techniques for the treatment of diabetes illustrate the transformative effects of AI in medicine to enhance treatment and improve quality of life.

**Author Contributions:** Conceptualization, A.C., K.S. and M.M.R.; methodology, A.C., K.S. and M.M.R.; software, M.R.A. and C.C.; validation, M.R.A., C.C. and Y.L.; formal analysis, M.R.A., M.M.R., C.C. and Y.L.; investigation, M.R.A., M.M.R., C.C. and Y.L.; resources, A.C., K.S. and M.M.R.; data curation, M.R.A. and M.M.R.; writing—original draft preparation, A.C., K.S. and M.M.R.; writing—review and editing, A.C., K.S., M.R.A. and M.M.R.; visualization, M.R.A.; supervision A.C., K.S. and M.M.R.; project administration, A.C.; funding acquisition, A.C., K.S. and M.M.R. All authors have read and agreed to the published version of the manuscript.

**Funding:** This research was funded by National Institutes of Health (National Institute of Diabetes and Digestive and Kidney Diseases) grant DP3 DK101075 and JDRF grants 1-SRA-2019-819-S-B and 2-SRA-2017-506-M-B made possible through collaboration between JDRF and The Leona M. and Harry B. Helmsley Charitable Trust).

**Institutional Review Board Statement:** The dataset used study in "ML with Free-Living Data for Digital Health Technologies in T1D" was donated by individuals with Type 1 diabetes to Tidepool. Its use at Illinois Institute of Technology was reviewed by the IRB of Illinois Institute of Technology (IRB-2019-057, Exempt).

**Informed Consent Statement:** Not applicable.

**Data Availability Statement:** Not applicable.

**Conflicts of Interest:** The authors declare no conflict of interest.

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
