# Peer review of "Artificial Intelligence Algorithms for Treatment of Diabetes"

_algorithms, doi:10.3390/a15090299_

Round 1

Reviewer 1 Report

The authors present in the abstract several points they want to address in their manuscript. However, data privacy and derivation of information from the EHR is not elaborated in the text. The overall description is rather general, without detailed information on used algorithms and data.

In the end of the introduction, the authors refer only to their own work. The state-of-art part should also mention work of others. In particular, diabetes and insulin dosing has been a hot t opic over many years. Application of machine learning in this area is covered by many articles and papers.

Section 2

Line 192: two mainstream fairness databases (ADULT, COMPAS) are mentioned, but without any reference.

There is no detailed information about the Tidepool dataset. It is not clear whether the information about the metabolism of each person is available, whether interpersonal variability was studied. It is also not clear whether the Tidepool database is publicly available.

Complete feature list is missing.

How many patient records were used?

There are many machine learning algorithms available. Choice of RNN application must be explained. The algorithm should be also compared with another algorithms (of a different type, not NN).

From the presented description it would not be possible to reconstruct the used algorithm.

Section 3

First and second generation of AID systems are mentioned, but without any reference.

Line 424: term "virtual subject" is introduced. It is not clear how the data was generated/simulated. It needs more detailed description.

Section 4 Conclusions

This section is very brief and general. It does not bring any new information.

The manuscript has 43 references in total. However, 65% of them are self-citations. It seems that the authors are not aware of research performed by other research groups around the world.

I recommend the authors to focus on one topic and go deeper in it. They should not re-use too many of their previously published results.

Reviewer 2 Report

Dear authors,

the manuscript is moderately interesting and in need of major revisions. Here are some indications.

- Please structure the abstract better, explain the methodology and the results better.

- The introduction from lines 27 to 112 has no bibliography, it would be appropriate to justify the statements that the authors make by supporting it with bibliography

- In Introduction from line 113 to line 118 11 studies are cited in five lines, it is excessive because it is very dispersed, it would be advisable to insert a few precise and punctual studies, or to keep the 11 studies but expand the introduction.

- Please structure the manuscript in: Introduction, Materials and Methods, Results, Discussion, Conclusions (optional).

- In Methods, please describe better the setting, the participants, the data sources.

- Discussion section is absent, please create and implement it.

- The conclusions are quite trivial and should be expanded by explaining also how they can be useful to the political decision-maker and how a possible public health intervention can improve the situation.

- Please check your English.

- The figures are adequate.

Please answer point by point.

Kind regards

Reviewer 3 Report

In this paper, the authors present a study to improve automated insulin delivery systems using machine learning approaches. This research has its merits and presents a good work; however, it needs major effort in order to be considered for publication. In this way, the authors have to do some changes in order to improve the publication.  The main directions for improvement are:  

·       The introduction section needs to be improved because citations are required for many claims made in this section, for example:

o   “The data used can be subjective, objective, or a combination of both…”, In which research are the data classified into these two types?

o   “Diabetes, a chronic disease that affects one out of every eleven people around the world will be used to illustrate ML applications for detection, classification, and prediction problems” who said that? World Health Organization? A citation is required here, please

o   “While the cure of diabetes is the objective of many active research programs, there is no cure approved to date for implementation.” Please indicate which are these research programs

·       At the end of the introduction, it is necessary to add a paragraph explaining the structure of the article and what readers will see in the following sections.

·       I miss a background section where the authors discuss about more recent works related to AI and Machine Learning applied to the health.

·       Section 2 is empty. It is needed to write some paragraphs explaining what the reader will see in this section.

·       I miss an evaluation and discussion section, where the authors explain better how the experiment was carried out and the main results obtained. This is not clear in the manuscript.

·       It would be interesting that the authors made a comparison of their results with the results obtained by more similar studies cases. Also, they have to explain why their approach is better than others. 

·       The conclusion section should be improved, explaining more in details the results obtained.  

·       It is needed to include a future works section to explain which the next research lines are to follow or include this information in the conclusion section and renaming it to “Conclusion and Future word”. 

Round 2

Reviewer 2 Report

The authors implemented the manuscript as suggested. Kind regards.

Reviewer 3 Report

Dear editor,

The authors have made the changes requested in the first revision, so, I believe the article can be considered for publication. Congratulations to the authors for the work done. 

Best regards